# Stepwise visualization of membrane pore formation by suilysin, a bacterial cholesterol-dependent cytolysin

**Carl Leung**[1†], **Natalya V Dudkina**[2,3†], **Natalya Lukoyanova**[2,3], **Adrian W Hodel**[1], **Irene Farabella**[2,3], **Arun P Pandurangan**[2,3], **Nasrin Jahan**[4], **Mafalda Pires Damaso**[4], **Dino Osmanović**[1,5], **Cyril F Reboul**[6], **Michelle A Dunstone**[6,7], **Peter W Andrew**[4], **Rana Lonnen**[4], **Maya Topf**[2,3], **Helen R Saibil**[2,3*‡], **Bart W Hoogenboom**[1,5*‡]

[1]London Centre for Nanotechnology, University College London, London, United Kingdom; [2]Department of Crystallography, Birkbeck College, London, United Kingdom; [3]Institute of Structural and Molecular Biology, Birkbeck College, London, United Kingdom; [4]Department of Infection, Immunity, and Inflammation, University of Leicester, Leicester, United Kingdom; [5]Department of Physics and Astronomy, University College London, London, United Kingdom; [6]Department of Biochemistry and Molecular Biology, Monash University, Melbourne, Australia; [7]Department of Microbiology, Monash University, Melbourne, Australia

**\*For correspondence:** h.saibil@mail.cryst.bbk.ac.uk (HRS); b.hoogenboom@ucl.ac.uk (BWH)

[†]These authors contributed equally as first authors to this work

[‡]These authors contributed equally as last authors to this work

**Competing interests:** The authors declare that no competing interests exist.

**Reviewing editor**: Volker Dötsch, Goethe University, Germany

**Abstract** Membrane attack complex/perforin/cholesterol-dependent cytolysin (MACPF/CDC) proteins constitute a major superfamily of pore-forming proteins that act as bacterial virulence factors and effectors in immune defence. Upon binding to the membrane, they convert from the soluble monomeric form to oligomeric, membrane-inserted pores. Using real-time atomic force microscopy (AFM), electron microscopy (EM), and atomic structure fitting, we have mapped the structure and assembly pathways of a bacterial CDC in unprecedented detail and accuracy, focussing on suilysin from *Streptococcus suis*. We show that suilysin assembly is a noncooperative process that is terminated before the protein inserts into the membrane. The resulting ring-shaped pores and kinetically trapped arc-shaped assemblies are all seen to perforate the membrane, as also visible by the ejection of its lipids. Membrane insertion requires a concerted conformational change of the monomeric subunits, with a marked expansion in pore diameter due to large changes in subunit structure and packing.

## Introduction

The bacterial CDCs and ubiquitous MACPF proteins are expressed as soluble monomers but assemble on membranes to form large, oligomeric pores. They form two branches of the largest superfamily of pore-forming proteins. Proteins of this MACPF/CDC superfamily share a common core topology of a highly bent and twisted β-sheet flanked by two α-helical regions, though lacking any detectable sequence homology between the two branches (*Rosado et al., 2008*). Crystal structures of CDCs in their soluble, monomeric form (perfringolysin, *Rossjohn et al., 1997*; anthrolysin, *Bourdeau et al., 2009*; suilysin, *Xu et al., 2010*; listeriolysin, *Köster et al., 2014*) revealed extended, key-shaped molecules. Pore-forming domains 1 and 3 (see also below) are linked by a long thin β-sheet (domain 2) to an immunoglobulin fold domain (4) which can bind to the membrane via a tryptophan-rich loop. CDCs form heterogeneous rings and arcs (*Dang et al., 2005*; *Tilley et al., 2005*; *Sonnen et al., 2014*) on cholesterol-rich liposomes and lipid monolayers (for example, the CDC perfringolysin O hardly binds to membranes with <30% molar concentration of cholesterol, *Johnson et al., 2012*). Extensive biophysical

**eLife digest** Many disease-causing bacteria secrete toxic proteins that drill holes into our cells to kill them. Cholesterol-dependent cytolysins (CDCs) are a family of such toxins, and are produced by bacteria that cause pneumonia, meningitis, and septicaemia.

The bacteria release CDC toxins as single protein molecules, which can bind to the membrane that surrounds the host cell. After binding to the membrane, the toxin molecules assemble in rings to form large pores in the host membrane. There are several stages to this process, but our understanding of what happens at the molecular level is incomplete.

Leung et al. studied suilysin, a CDC toxin produced by a bacterium that has a big impact on the pig farming industry because it causes meningitis in piglets. The bacterium can also cause serious diseases in humans through exposure to contaminated pigs or pig meat.

Leung et al. used a technique called electron microscopy to obtain atomic-scale snapshots of the toxin structures before and after the toxins were inserted into the membrane. In addition, real-time movies of the process were gathered using another technique called atomic force microscopy.

The experiments show that suilysin forms assemblies on the membrane that grow by one molecule at a time, rather than by the merging of larger assemblies of molecules. This results in a mixture of ring-shaped and arc-shaped toxin assemblies on the membrane. The arcs of suilysin are incomplete ring assemblies, but they are still able to make holes in the cell membrane. In order to insert into the membrane, the toxin molecules in the arcs and rings undergo a dramatic change in shape.

Understanding suilysin how CDCs assemble in membranes will guide further work into the development of new vaccines that can target these proteins to reduce the damage caused by bacterial infections.

and molecular analysis of CDCs established that, on CDC binding to the membrane (*Ramachandran et al., 2004*; *Hotze et al., 2012*), α-helical regions in domain 3 unfurl to form transmembrane β-hairpins, denoted as TMH1 and TMH2 (*Shepard et al., 1998*; *Shatursky et al., 1999*). If the TMH regions are trapped by introducing a disulphide bond (*Hotze et al., 2001*), prepore oligomers are formed on the membrane surface. Cryo-EM and single particle analysis of liposome-bound CDCs led to low-resolution 3D structures of prepore and pore forms of pneumolysin, a major virulence factor of *Streptococcus pneumoniae* (*Tilley et al., 2005*). These structures, as well as an AFM study of perfringolysin (*Czajkowsky et al., 2004*), established that the 11 nm high molecule must collapse to a height of 7 nm above the membrane in order to insert the TMH regions. Simple pseudo-atomic models were obtained by fitting domains (broken at plausible hinge points) into the EM density maps. It was proposed that the long, thin β-sheet domain 2 collapses after the molecule opens up to release the TMH regions. However, because of the heterogeneity of the oligomeric assemblies and aggregation of the liposomes upon pore formation, resolution has been limited by the difficulty of obtaining sufficiently large data sets.

After comparing several CDCs (pneumolysin, suilysin, anthrolysin, and listeriolysin), we found that suilysin was less susceptible to these problems and we chose it to pursue new structural and dynamic studies. A disulphide-locked double cysteine mutant of suilysin, designed to prevent TMH1 insertion, enabled us to trap an active prepore state as well as to visualize the pore formation process by AFM in solution. Cryo-EM reconstruction and fitting revealed new details of the β-sheet unbending and changes in subunit packing upon conversion of prepores to pores. AFM images reveal that the pre-pore state is highly mobile. Following the addition of DTT to trigger insertion of the disulphide-locked prepore, time-lapse AFM yielded real-time movies of its conversion to ring and crescent-shaped pores. The observed distributions of rings and arcs can be explained by a theoretical model for kinetically trapped, noncooperative assembly, fully determined by the relative kinetics of monomer binding to the membrane and monomer assembly on the membrane surface. Together these studies provide substantial new understanding of the structure and dynamics of CDC pore formation.

## Results

### Conformational changes in prepore and pore states determined by cryo-EM

Negative stain EM and rotational symmetry analysis of complete rings of disulphide locked (Gly52Cys/Ser187Cys) suilysin prepores and wild-type pores formed on lipid monolayers revealed that most rings

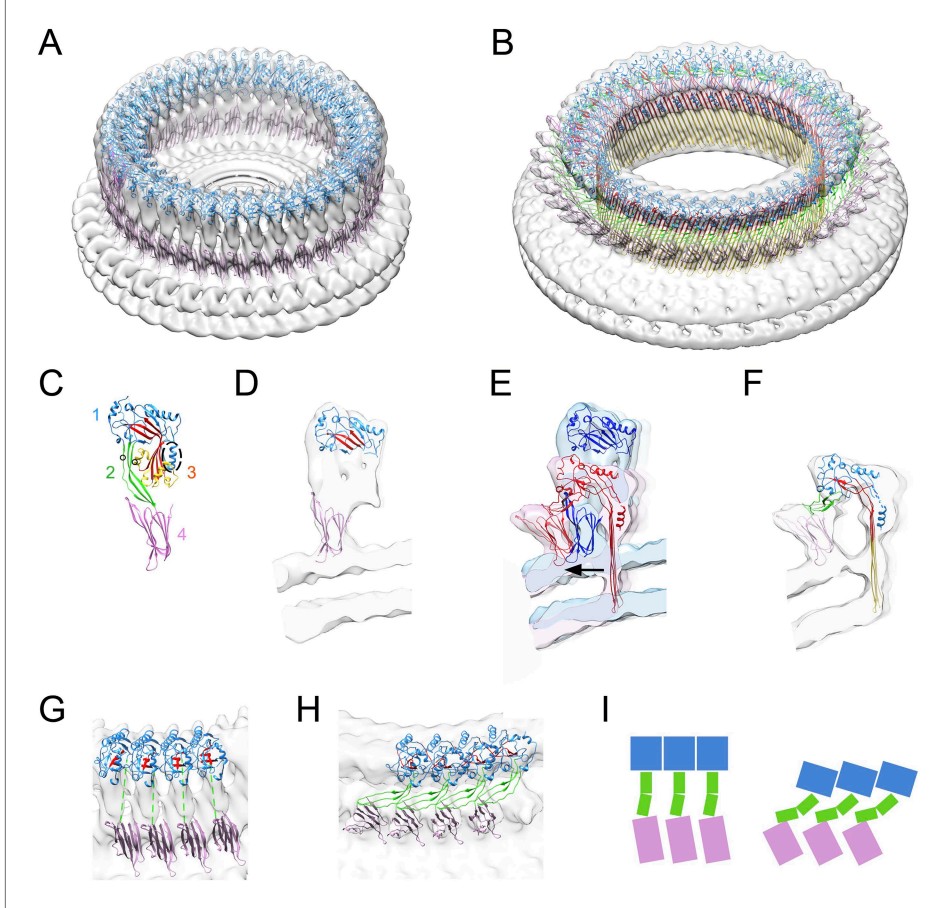

**Figure 1**. Structural transitions during pore formation. 3D cryo-EM maps of 37-mer prepore and pore forms of suilysin are shown with fitted atomic structures. (**A**) Density map of prepore, surrounded by the extracted disk of membrane, with domains 1 and 4 fitted. (**B**) Density map of pore with all domains fitted, including the β-barrel with strands at 20° tilt. (**C**) Suilysin crystal structure with the domains labelled, showing positions of cysteines introduced in the locked form (black circles) and a helical domain adjacent to the bend in the central β-sheet (dashed oval). (**D**) Cross-section through one side of the prepore map with the partial fit of atomic structures. (**E**) Overlay of one side of the prepore (blue) and pore maps (red), aligned to the same centre, showing the displacement of domain 4 (arrow). (**F**) Pore section with fit. (**G**) View of 4 subunits from outside the prepore. (**H**) View of 4 subunits from outside the pore. (**I**) Cartoons of domain packing in prepore and pore.

The following figure supplements are available for figure 1:

**Figure supplement 1**. Symmetry of suilysin prepores and pores, determined by negative-stain EM.

**Figure supplement 2**. Resolution curves for EM maps.

**Figure supplement 3**. Electrostatic potential maps and interacting residues.

**Figure supplement 4**. Comparison between prepore and crystal structure conformations.

---

contain 37 subunits (*Figure 1—figure supplement 1*). Unexpectedly, the diameter of the 37-fold suilysin prepore was smaller than the diameter of the 37-fold pore (see below for quantification), indicating that conformational changes during pore formation are accompanied by changes in subunit packing.

3D reconstruction of suilysin prepores and pores in liposomes was performed using a pseudo single-particle approach (*Tilley et al., 2005*), yielding a 15 Å cryo-EM map of the prepore using the disulphide-locked construct (*Figure 1A,D*), and a 15 Å cryo-EM map of a wild-type suilysin pore (*Figure 1B,F*, see also *Figure 1—figure supplement 2*). The 3D maps, both of 37-mers, confirmed a

significant expansion in ring diameter upon pore formation. In order to interpret the maps, we performed flexible fitting of suilysin domains from the crystal structure (*Figure 1C*; *Xu et al., 2010*; PDB:3hvn). Domain deformations and hinge movements were identified by normal mode analysis (*Lindahl et al., 2006*). Additional evidence for the correctness of the fits followed from electrostatic potential maps and analysis of interacting residues at the interfaces of domain 1 (*Figure 1—figure supplement 3*). The results show that both models have extended regions of complementary charge on the predicted interacting surfaces. Although the extent of complementary charge is less in the pore model, in this case the oligomer is stabilized by the β-barrel of the pore. Measured from the fitted position of the base of domain 4, the prepore and pore diameters are 296 Å and 319 Å, respectively (*Figure 1E*), an expansion of 8%. In addition to the 4 nm reduction in height, each pore subunit approximately doubles in width.

The prepore is distorted from the crystal structure, with some collapse of domain 2 and opening of the β-sheet (*Figure 1—figure supplement 4*), despite the presence of the disulphide bridge (*Figure 1D*). However, the map features are not sufficiently defined to guide fitting, most likely owing to the greater flexibility of the prepore state, as described below.

The pore structure is similar to that observed with pneumolysin, but with improved resolution, and showing significant differences in the hinge bending and domain movements. The β-strands are tilted by 20°, in agreement with previous results (*Reboul et al., 2012*; *Sato et al., 2013*). The distortion to domain 2 differs from that proposed in the earlier model (*Tilley et al., 2005*). Seen from outside the ring, domain 2 collapses sideways, to the right, such that domain 1 is aligned above the adjacent domain 4, with a sideways tilt that expands the ring. The expansion is clearly seen in the wider spacing between subunits at domain 4 (*Figure 1G,H,I*). Domain 2 must bend at a central hinge point to fit into the EM density (*Figure 1H,I*; *Reboul et al., 2014*). As expected, there is a major opening of the bent β-sheet. In addition, a helical subdomain flanking the bend of the central β-sheet (residues 335–347, dashed oval in domain 3, *Figure 1C*) moves as a separate rigid body, as also shown by a spectroscopic study (*Ramachandran et al., 2004*). Notably, the equivalent region has been implicated in the triggering mechanism for unbending in a recent EM study of a remotely related MACPF protein (*Lukoyanova et al., in press*).

## Real-time visualization of the prepore-to-pore transition and membrane perforation

When imaged by negative-stain EM, both the prepore and pore states appeared in heterogeneous ring- and arc-shaped assemblies (*Figure 2A,B*; *Sonnen et al., 2014*; *Köster et al., 2014*). The expansion in ring diameter upon membrane insertion was confirmed by a statistical analysis of the radius of curvature of the arc assemblies (*Figure 2—figure supplement 1*), which also revealed significantly larger variations in arc curvature, that is, larger flexibility, for the prepore than for the pore state.

To facilitate AFM imaging of the disulphide-locked suilysin prepores, the protein was confined to well-defined domains on phase-separated lipid membranes (*Connell et al., 2013*), showing densely packed suilysin rings and arcs that extended 10–11 nm above the membrane surface (*Figure 2C,D*), consistent with the structural data for the prepore state (*Figure 1A,D*).

At lower packing density on the membrane, suilysin prepores were only resolved when the temperature was lowered to 15°C. Cooling appeared to reduce the prepore mobility such that individual prepore assemblies could be observed while diffusing over the membrane (*Video 1*). When imaged at room temperature, suilysin prepores appeared as streaks in the AFM images—as can be expected for highly mobile proteins—with slightly improved contrast at the lipid phase boundaries (*Figure 3A*). As demonstrated by real-time AFM images of the same area on the membrane, the disulphide-locked suilysin reproducibly converted from the prepore to the pore state upon exposure to DTT (*Figure 3A–E*, *Figure 3—figure supplement 1*): in less than a minute, exposure to DTT triggered the appearance of diffuse rings and arcs that became progressively clearer and more prominent (bottom half of *Figure 3A*), adopting the reduced height typical of the pore state (*Figure 3B–C*). This process was accompanied by a gradual disappearance of the diffuse streaks (i.e., suilysin prepores), while additional high (white) features appeared on the surface. We identify these features as lipid micelles or fragments being ejected from the membrane. This interpretation is supported by the subsequent appearance of larger plateaus that were consistent in height with the collapse of newly formed lipid layers on top of the membrane (*Figure 3D*). After about 20 min, the membrane was cleared of these features, leaving a heterogeneous population of suilysin pore assemblies perforating the membrane

**Figure 2**. Negative-stain EM and AFM of disulphide-locked suilysin. (**A**) Negative-stain EM disulphide-locked suilysin (ds-SLY) on egg PC:cholesterol monolayers (45:55%), locked in the prepore state (−DTT). (**B**) as (**A**), for disulphide-locked suilysin incubated in the presence of 5 mM DTT in solution to reduce the disulphide bridge, so that the suilysin is rapidly converted to the pore conformation. (**C**) AFM of densely packed suilysin prepores, confined to the egg PC-rich domain of a phase-separated egg PC:DDAB:Cholesterol (33:33:33%) supported lipid bilayer, with its corresponding height distribution (**D**) referenced to the membrane surface.

The following figure supplement is available for figure 2:

**Figure supplement 1**. Radius of curvature for arc-shaped suilysin assemblies in the prepore and pore states.

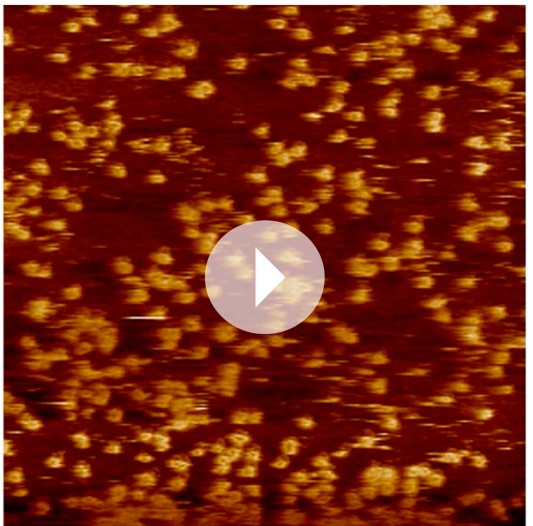

**Video 1**. Mobile disulphide-locked suilysin (prepore) assemblies diffusing on the membrane. At a temperature of 15°C, the mobility of disulphide-locked suilysin is sufficiently reduced for the assemblies to be resolved by real-time AFM at 15 s/frame. This sequence of images was captured ~30 min after protein injection and at 384 pixels per line. The timing of the video is accelerated by a factor of ~100. Full z-colour scale = 20 nm.

(**Figure 3E**). The transition from prepore to pore, as well as the emergence and clearance of lipid aggregates, could also be observed via height profiles taken along various topographic features in these images (**Figure 3F,G**).

## AFM demonstrates that incomplete, arc-shaped assemblies perforate the membrane

The heterogeneity of ring- and arc-shaped assemblies was confirmed for wild-type suilysin by negative-stain EM on egg PC:cholesterol monolayers and by AFM in solution on supported bilayers (**Figure 4A,B**). EM and AFM yielded quantitatively similar arc-length distributions for identical lipid composition, protein concentration, and incubation temperature (**Figure 4—figure supplement 1**). Qualitatively similar behaviour could be observed by negative-stain EM on lipid vesicles (**Figure 4A**, inset). Unlike the disulphide locked construct, wild-type suilysin was converted from its soluble, monomeric state (**Figure 4—figure supplement 2**) to the pore state without the appearance of prepore intermediates, within the time resolution of our AFM experiments. Rings and arcs had a height of 7–8 nm in AFM (**Figure 4B**, inset), in agreement with the cryo-EM data on the pore state (**Figure 1B,F**).

To verify if suilysin rings and arcs perforate the membrane, we used high-aspect ratio AFM tips to probe the membrane in the pore lumen (**Figure 3G**). Both rings (**Figure 4C**) and arcs (**Figure 4D**) enclosed local depressions in the membrane, the depth of which was tip-dependent, but in many cases exceeded the 2.2 nm length of an extended lipid molecule, indicating that the lipid bilayer was locally removed.

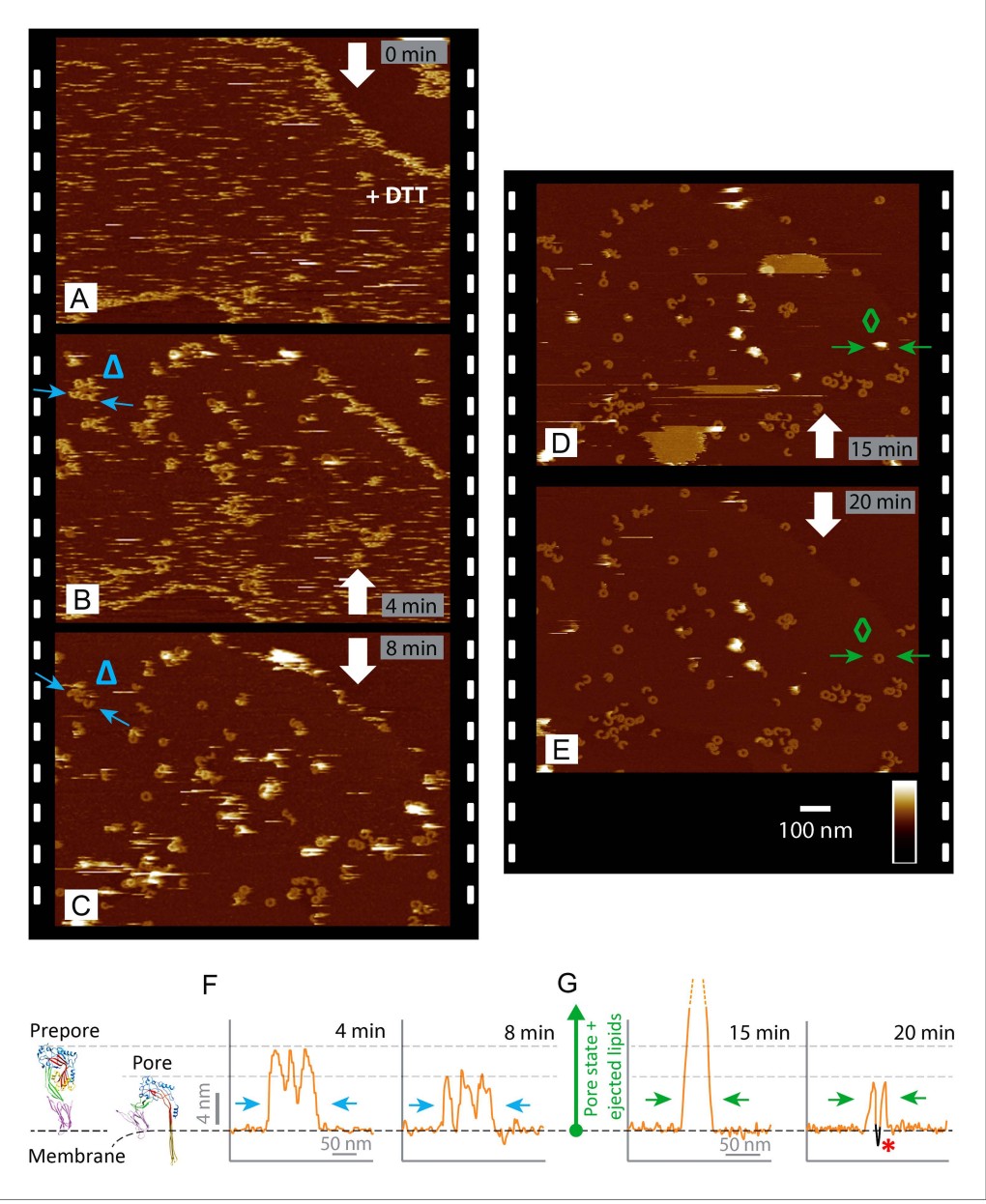

**Figure 3**. Real-time imaging of the prepore-to-pore transition and membrane perforation by suilysin. Subsequent AFM frames of the same area were alternatively recorded from top to bottom and from bottom to top, as indicated by white arrows. Frame time: 4 min, colour scale: 35 nm. (**A**) Loosely bound to sphingomyelin-rich domains in the phase-separated lipid mixture (DOPC:sphingomyelin:cholesterol, 33:33:33%), the prepore intermediates of disulphide-locked suilysin appear as diffuse streaks. 5 mM of DTT is injected on about 50% completion of the scan. (**B**) On consecutive scanning, the streaks become more clearly defined as arc-shaped oligomers and complete rings. Towards the top end of the scan, clusters of arc-shaped complexes, mostly in the prepore intermediate (~10.5 nm high), can be distinguished (Δ). (**C**) With the scan direction reversed, and the same area scanned again, the cluster of prepore complexes has converted into the pore state (~7.5 nm high), within ~2 min. The prepore to pore transition is followed by the ejection of globular features of varying dimensions exceeding 15 nm above the suilysin in the pore state. We interpret these as ejected lipids. (**D**) These lipids gradually detach from the surface on the pore state suilysin assemblies and can be observed as patches of lipids condensing back onto the membrane. The prepore to pore transition of the suilysin is now complete. (**E**) After ~20 min, the surface is almost clear of the ejected lipids. (**F**) Cross-sectional line profile extracted as indicated (Δ in **B**–**C**), illustrating the prepore to pore

*Figure 3. Continued on next page*

*Figure 3. Continued*

transition. (**G**) Cross-sectional line profile extracted as indicated (◊ in **D**–**E**), illustrating the lipid ejection and eventual formation of an aqueous pore in the membrane (*).
The following figure supplement is available for figure 3:

**Figure supplement 1**. Reproducibility of real-time imaging of the prepore-to-pore transition and membrane perforation by suilysin.

These depressions were reproducible between the trace and retrace versions of the AFM line scans and were observed for various lengths and orientations of the arcs (*Figure 4E*). We therefore conclude that suilysin can perforate the membrane irrespective of the completion of the ring assembly, locally removing the lipids to create partial β-barrel pores with an unsealed edge of the lipid bilayer. This is in agreement with recent cryo electron tomography data on pneumolysin assemblies (*Sonnen et al., 2014*).

Besides isolated rings and arcs, we observed interlocked arcs (*Figure 4A,B*, arrows), in which one or both ends of the arc contacted another arc or ring. As was the case for the isolated arcs, we found these interlocked arcs capable of perforating the membrane and form membrane lesions that can be smaller, but also larger than those in closed rings (*Figure 4F*). Once assembled in the pore conformation, the arcs were stable and did not evolve further; even interlocked arcs did not merge into complete rings (*Figure 4G*).

Interestingly, the prepore-to-pore transition and membrane perforation by the wild-type suilysin was largely prevented, in a dose-dependent manner, by adding the disulphide-locked mutant in the incubation process (*Figure 4—figure supplement 3*). Subsequent exposure to DTT restored normal pore formation.

## Suilysin assemblies are consistent with kinetically trapped oligomerization

To analyze the oligomerization process, we measured the arc-length distributions of wild-type suilysin in the pore state and of the disulphide-locked mutant in prepore (−DTT) and reduced, pore (+DTT) configurations, based on negative stain EM analysis under the same conditions (*Figure 5A–C*). The distributions show a broad peak centred between lengths of 10–30 monomers and a smaller, narrow peak corresponding to completed rings of about 37 monomers. The arc-length distributions for disulphide-locked prepores and pores after reduction by DTT are practically identical. Combined with the observation that suilysin does not oligomerize before binding to the cholesterol-containing membrane (*Figure 4—figure supplement 2*), this demonstrates that assembly is completely determined and terminated in the prepore state, i.e., is not affected by the prepore-to-pore transition. This conclusion is further confirmed by the lack of growth of individual arcs in the pore state upon subsequent, further addition of wild-type suilysin (*Figure 5—figure supplement 1*), and greatly simplifies the interpretation of the arc-length distributions.

To explain these distributions, we calculated the oligomeric populations for a model in which monomers from the solution irreversibly bind to the membrane with a rate constant $k_b$, and in which irreversible oligomerization on the membrane occurs only by monomer addition, with a rate constant $k_a$ (*Figure 5D*). In such a simple model, the oligomerization reaction is arrested by depletion of monomers, yielding kinetically trapped assembly intermediates. Since the resulting populations only depend on the ratio $k_a/k_b$ and on the (experimentally known) total number of monomers per unit area of membrane (*C*), this model can be used to fit the experimental data with a single free parameter ($k_a/k_b$). The broad peak for intermediate arc-lengths (*Figure 5A–C* and *Figure 4—figure supplement 1*) can thus be explained by a ratio $k_a/k_b$ that is sufficiently large to ensure a steady supply of monomers to sustain the oligomerization reaction, but not large enough to yield only completed assemblies (i.e., rings).

## Discussion

CDCs are protein toxins that are potent virulence factors in bacteria. They are part of the major MACPF/CDC superfamily of pore-forming proteins. Our data map the structure (*Figure 1*) and assembly pathways of membrane pore formation by the CDCs in unprecedented detail and accuracy, as summarized in *Figure 6*. Using the disulphide-locked suilysin variant, we have resolved the

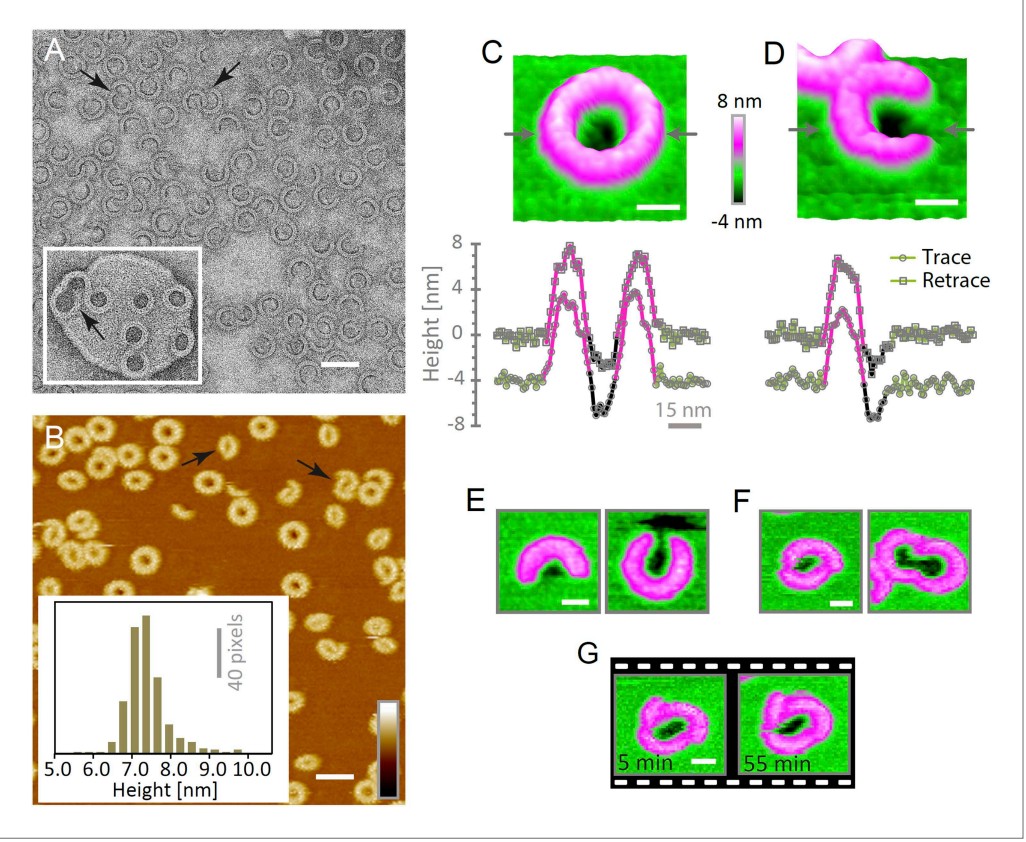

**Figure 4**. Suilysin assembles into ring- and arc-shaped oligomers that perforate the membrane. (**A**) Negatively stained EM of arc- and ring-shaped assemblies of wild-type suilysin on an egg PC:cholesterol (45:55%) lipid monolayer, and (inset) on a liposome of egg PC:cholesterol (45:55%). (**B**) AFM topography of wild-type suilysin on a supported egg PC:cholesterol (67:33%) lipid bilayer. The wild-type suilysin extends 7–8 nm above the lipid bilayer background, as indicated by the height histogram for 402 individual particles (inset). (**C**) The AFM topography of a complete suilysin ring reveals a circular hole (dark) in its lumen, whereas the lipid bilayer surrounding the ring remains intact (green). (**D**) The topography of a suilysin arc shows a hole (dark) in the membrane only partially enclosed by the suilysin assembly. Images in **C** and **D** are shown in a 15° tilted representation, and height profiles measured across the ring/arc confirm membrane perforation. (**E**) Examples of wild-type suilysin arcs of different lengths. Transmembrane holes are consistently observed. (**F**) Examples of interlocked-arc assemblies. As shown in the right image, the membrane area removed by the two arcs is larger than the hole in the complete ring (**C**). (**G**) Sequence of AFM images of the same interlocked-arc assembly, stable for at least 50 min. Scale bars **A**–**B**: 50 nm, **C**–**G**: 15 nm, full z colour scale **B**–**G**: 12 nm.

The following figure supplements are available for figure 4:

**Figure supplement 1**. Suilysin pore assemblies by EM and AFM.

**Figure supplement 2**. Suilysin is a monomer in solution.

**Figure supplement 3**. AFM assays of wild-type suilysin (WT-SLY) doped with disulphide-locked suilysin (ds-SLY).

initial membrane binding of suilysin monomers (*Figure 4—figure supplement 2*), oligomerization (*Figure 5*), and membrane insertion stages (*Figures 2–4*) in pore formation.

Suilysin oligomers exhibit a broad distribution of arc- and ring-shaped assemblies (*Figures 4 and 5*). Under given conditions such as incubation temperature, lipid composition, protein sequence and concentration, these distributions are reproducible between EM and AFM experiments (*Figure 4—figure supplement 1*). The similarity between prepore and pore distributions (*Figure 5B,C*) implies that oligomerization is arrested before pore insertion. Therefore, our work rigorously establishes that the

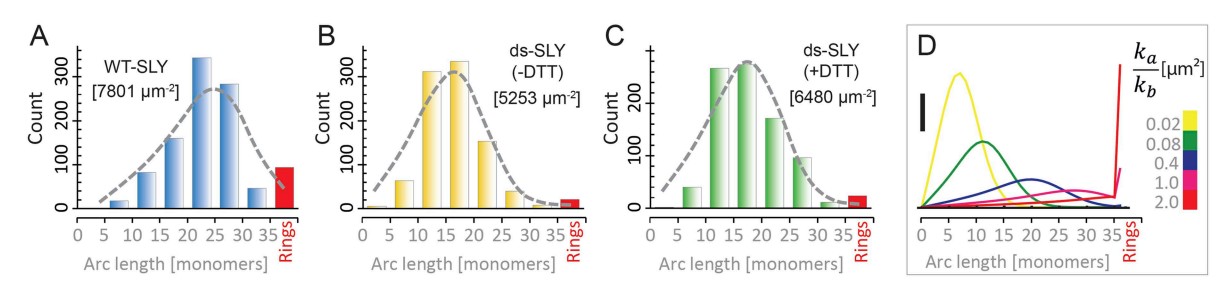

**Figure 5**. Oligomerization states for arc- and ring-shaped assemblies of suilysin. (**A**) The arc-length distribution of wild-type suilysin displays a broad peak for arcs that contain between 15 and 30 monomers, and a smaller, sharp peak for complete rings (37-mers). (**B**) Arc-length distribution for the disulphide-locked suilysin prepore intermediate. (**C**) For the disulphide-locked mutant incubated in the presence of DTT (pore-state), the arc-length distribution is practically identical to the distribution for the prepore-locked intermediate. (**D**) Calculated arc-length distributions for a simple model of kinetically trapped oligomerization, with $C$ = 2000 monomers per square micron (see 'Materials and methods'). The peak of the arc-length distribution shifts from smaller to larger oligomers on increasing the ratio between the rate constants for monomer association ($k_a$) and monomer binding to the membrane ($k_b$). Vertical scale bar: 40 counts. Grey, dashed lines in **A**–**C** denote fits of the experimental data with the oligomerization model, yielding $k_a/k_b$ = 0.893 ± 0.008 µm$^2$ (**A**); 0.438 ± 0.012 µm$^2$ (**B**); 0.425 ± 0.012 µm$^2$ (**C**). Numbers in brackets in **A**–**C** indicate the estimated total number of monomers per square micron. The experimental data here are based on negative-stain EM images on monolayers of egg PC:cholesterol (45:55), incubated at 37°C.

The following figure supplements are available for figure 5:

**Figure supplement 1**. Sequential addition of wild-type suilysin in the pore state.

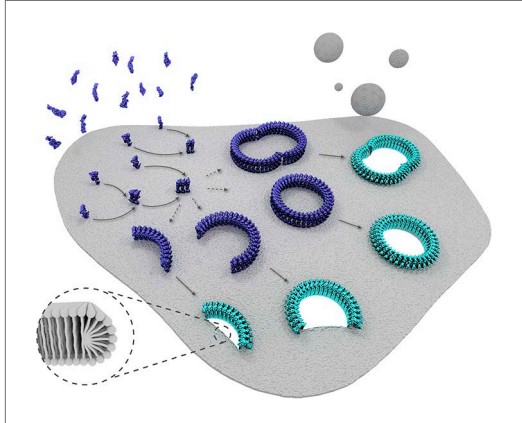

**Figure 6**. Schematic representation of suilysin membrane binding, assembly, and pore formation. From left to right: monomers bind to the membrane and oligomerize. The assembly of monomers proceeds in the prepore intermediate and results in either complete rings or kinetically trapped arc-shaped oligomers. The arc- and ring-shaped assemblies subsequently collapse to the pore configuration with the transmembrane β-hairpins unfurled and inserted into the lipid bilayer in a concerted conformational change. Lipids are subsequently ejected from the membrane (shown as grey spheres) and aqueous pores of different sizes are formed in the membrane. The inset shows a possible configuration of lipids at the unsealed edges of the bilayer.

whole CDC assembly takes place in the prepore state, as was previously suggested for perfringolysin (*Hotze et al., 2001*; *Hotze and Tweten, 2012*).

As can be deduced from the prevalence of larger but incomplete assemblies of the disulphide-locked suilysin, the association of such larger oligomers ($\gtrsim$ 5 subunits) is not a determining factor in membrane pore formation by CDCs. On the contrary, the oligomerization appears to be dominated by monomer addition (*Figure 5*), although the addition of smaller oligomers ($\lesssim$ 5 subunits) cannot be fully excluded. As demonstrated by our oligomerization model, the measured arc-length distributions are consistent with the kinetically trapped product of a two-stage irreversible and noncooperative reaction. The outcome is largely determined by the ratio of the corresponding rate constants and the density of monomers per unit area of membrane (in addition to the effects of steric hindrance at higher surface densities). The first stage can be the binding of monomers to the membrane, as assumed here, or a rate-limiting nucleation step that triggers the oligomerization reaction, as assumed elsewhere (*Hotze et al., 2001*). The second stage of this reaction is oligomerization by addition of monomers or very small oligomers. As the rate constants can be expected to vary from one protein to another, this model implies that the various CDCs can yield differing oligomeric populations.

Suilysin prepore intermediates appear both more mobile (*Figure 3A–E*) and more flexible than pore assemblies (*Figure 2—figure supplement 1*). This is consistent with the observation that the β-sandwich membrane-binding domain 4 does not significantly penetrate the membrane (*Nakamura et al., 1995*; *Ramachandran et al., 2002*). In contrast to the earlier observations on pneumolysin (*Tilley et al., 2005*), prepore assembly in suilysin is accompanied by some opening of the central β-sheet. A possible explanation for this difference is that the wild-type pneumolysin preparation in that study formed many stable prepores, apparently in an inactive, dead-end state, whereas wild-type suilysin is extremely active and is not observed in a prepore state.

The activity of wild-type suilysin was greatly impaired, however, by incubating it in the presence of an equal concentration of disulphide-locked suilysin (*Figure 4—figure supplement 3*) and recovered on unlocking the mutant. These results imply that wild-type and mutant co-assemble as expected, and that the prepore-to-pore transition requires a concerted conformational change of all subunits in the suilysin assemblies, suggesting a cooperative insertion of subunits into the membrane.

The mobility of prepore assemblies on the membrane surface makes it possible for the subunits to slide apart upon pore formation, as seen in the 8% diameter expansion. This size difference was seen in the earlier work, but the symmetry measurement was less clear, and it was explained by assigning a lower symmetry to the prepore (31 vs 38 for the pore) (*Tilley et al., 2005*). In view of the present observations, it seems likely that pneumolysin rings also expand and that the previous assignment of different symmetries to pneumolysin pores and prepores was most likely incorrect.

We observed arc-shaped assemblies as small as 5 subunits, with heights corresponding to the suilysin pore state. This gives an estimate of the minimum oligomer size required for membrane insertion. The size of the AFM tip was too large to probe the membrane perforation in arc-shaped complexes smaller than about 15 monomers (see e.g., *Figure 4E*). We have observed membrane perforation for arcs at any size between 15 and 37 monomers, which unambiguously demonstrates that a fully enclosed β-barrel is not essential for the insertion of the transmembrane β-hairpins and membrane perforation, as was previously presumed (*Hotze and Tweten, 2012*). Unsealed lipid edges and incomplete β-barrels are a surprising by-product of this membrane perforation by incomplete CDC oligomers (*Figure 4D,E*), as was recently suggested based on electron tomography of various pneumolysin assemblies (*Sonnen et al., 2014*).

The size of the membrane lesions can thus vary between less than half the lumen in a completed SLY pore to the larger pores formed by interlocked arcs (*Figure 4F,G*), as suggested by conductance measurements on black lipid membranes (*Marchioretto et al., 2013*).

Regardless of the extent of oligomerization, both arc and ring prepores can convert directly to the pore configuration as the β-hairpins unfurl and insert into the lipid bilayer. This is followed by the ejection of lipids from the membrane as the pore is formed (*Figure 3C,D* and *Figure 3—figure supplement 1C,D*). The results imply that the hydrophilic inner surface of the partially or fully completed β-barrel is sufficient to destabilize the lipid membrane in the pore lumen, leading to ejection of lipid micelles from the pore.

The approach of identifying hinge regions for domain fitting to the suilysin pore map has yielded a pseudo-atomic model that goes beyond the earlier rigid body fitting to pneumolysin (*Tilley et al., 2005*). The mechanism of collapse through buckling of domain 2 involves a sideways movement around the ring, and the bending is in the opposite direction to that proposed in the earlier model. The resulting tilt of domain 1 results in the 8% radial expansion of the ring, clearly seen by the displacement of domain 4 (*Figure 1E*). A helical subdomain thought to be involved in triggering of sheet opening in a MACPF protein (dashed oval in domain 3, *Figure 1C*; *Lukoyanova et al., in press*) is also likely to move in suilysin, strengthening the notion that the mechanism of unfolding and pore formation is conserved between the remotely related MACPF and CDC subfamilies.

In summary, we have visualized the various stages of membrane pore formation by a CDC at greatly improved spatial and temporal resolution, to provide new insights in domain movements and pathways of assembly for a major superfamily of pore-forming proteins.

## Materials and methods

### SLY expression and purification

#### Cloning of wild-type suilysin gene

Genomic DNA was extracted from *S. suis* (kindly provided by Dr Vanessa Terra, London School of Hygiene and Tropical Medicine; UK), using the PureLink genomic DNA mini kit (Invitrogen, Carlsbad, CA).

The suilysin gene was amplified using the primers: SLY_F, 5'- CGG CGC CAT GGC TTC CAA ACA AGA TAT TAA TCA GTA TTT TCA AAG -3' and SLY_R, 5'-GAT AGG ATC CTC ACT CTA TCA CCT CAT CCG CAT ACT GTG-3'. These primers introduced restriction sites for NcoI and BamHI for subsequent cloning into pEHISTEV (*Liu and Naismith, 2009*), in-frame with and downstream of nucleotides encoding a 6-histidine tag and a TEV protease cleavage site. The recombinant plasmid expressing suilysin was then transformed into *Escherichia coli* XL-10 Gold competent cells (Agilent, Santa Clara, CA), according to the manufacturer's instructions. Positive clones were identified by blue-white colony screening and were confirmed by sequencing. Plasmid carrying the suilysin gene was subsequently transformed into *E. coli* Rosetta-2 (DE3) (Novagen, Millipore, Watford, United Kingdom).

## Creation of a disulphide-locked construct of suilysin

This mutant was made using a QuikChange Multi Site-Directed Mutagenesis Kit (Agilent). The primers G52C_F, 5'-CAC AAG AGA TTC TTA CAA ATG AGT GCG AAT ACA TTG ATA ATC CGC CAG C-3' and G52C_R, 5'-GCT GGC GGA TTA TCA ATG TAT TCG CAC TCA TTT GTA AGA ATC TCT TGT G-3' were used to replace the codon for Gly52 (GGA) in the wild-type with a Cys codon (TGC). The primers S187C_F, 5'-TGA AAC AAT GGC ATA CAG TAT GTG CCA ATT GAA AAC GAA GTT CGG AAC-3' and S187C_R, 5'-GTT CCG AAC TTC GTT TTC AAT TGG CAC ATA CTG TAT GCC ATT GTT TCA-3' were used to substitute the Ser187 codon (TCA) by a Cys codon (TGC) in the same construct. Following the digestion of the methylated/hemimethylated parental DNA with DpnI, the construct containing the double mutation was transformed into *E. coli* XL-10 Gold (Agilent). Positive clones were identified by blue–white colony screening and mutations confirmed by sequencing. Purified plasmid from selected clones was subsequently transformed into *E. coli* Rosetta-2 (DE3) (Novagen).

## Expression and purification

Recombinant wild-type suilysin and the *cys*-locked version were expressed in Overnight ExpressTM Instant TB medium (Novagen) containing kanamycin (50 µg/ml; Sigma–Aldrich, Dorset, United Kingdom). Cells were harvested by centrifugation and lysed with 1x BugBuster protein extraction reagent (Novagen) supplemented with 10 µg/ml DNase I (Sigma–Aldrich), 5 mM MgCl$_2$, and EDTA-free protease inhibitor (Roche Applied Science, Welwyn Garden City, United Kingdom). Soluble cellular extracts were clarified and loaded onto a HisTrap High Performance column (GE Healthcare, Little Chalfont, United Kingdom) in loading buffer (20 mM Tris–HCl, 150 mM NaCl, 20 mM Imidazole, pH 7.5). The column was washed thoroughly with 20 mM Tris–HCl, 150 mM NaCl, 50 mM Imidazole, pH 7.5, and the 6-histidine-tagged wild-type and cys-locked suilysins were eluted using a stepwise gradient of elution buffer (20 mM Tris–HCl, 150 mM NaCl, containing 0 to 500 mM Imidazole, pH 7.5). The purity of the eluted fractions was assessed by SDS-PAGE (*Laemmli, 1970*). The haemolytic activity of the suilysins was determined as described previously (*Owen et al., 1994*), except that 5 mM DTT was included in some assays as appropriate. Wild-type suilysin had a specific activity of 39,000 HU (haemolysis units)/mg protein. No haemolytic activity was seen with the cys-locked construct, in the absence of DTT, but in 5 mM DTT, its specific activity was 20,000 HU/mg protein.

## Lipid and liposome preparation

All lipid materials (cholesterol, egg PC, DOPC, sphingomyelin, DDAB) were purchased from Avanti Polar Lipids (Alabaster, AL). Small unilamellar lipid vesicles were prepared by the extrusion method (*Hope et al., 1985*). Briefly, lipids in powdered form were weighed and dissolved in chloroform to produce a homogeneous mixture with a lipid concentration of ~1 mg/ml. The solvent was then slowly evaporated for at least 5 hr by passing a steady stream of argon in a fume hood, yielding a dry lipid film. The lipid film was resuspended, by vigorous vortexing for 5 min, in 1 ml of 20 mM Tris, 150 mM NaCl, pH 7.8, to form large, multilamellar vesicles. This solution was transferred to a Fisherbrand FB11201 bath sonicator (Fisher Scientific, Loughborough, UK), maintained above the gel–liquid transition temperature of the constituent lipids. The large multilamellar vesicles were disrupted by 15-min sonication treatments at frequencies between 40 and 80 kHz, interspersed by two freeze/thaw cycles. The solution containing the lipid dispersion was loaded into an Avanti mini-extruder kit (Avanti Polar Lipids) and kept above the transition temperature of the lipids. The lipid solution was forced through a Whatman Nucleopore polycarbonate filter (GE Healthcare Lifesciences, Buckinghamshire, UK) with an 80 nm nominal pore diameter. The extrusion process was repeated at least 30 times to yield small unilamellar vesicles with a diameter near the pore size of the filter used, as verified by negative stain EM.

For EM experiments, liposomes were prepared from 5 mM lipids containing ~45 mol% of egg PC and ~55 mol% cholesterol resuspended in 100 mM NaCl, 50 mM HEPES, pH 7.5 by extrusion through an 80-nm filter as previously described (*Tilley et al., 2005*).

## Electron microscopy sample preparation and data acquisition

### Negative stain

10 µg/ml monomeric wild-type suilysin was negatively stained with 2% wt/vol uranyl acetate. To form prepore and pore complexes on lipid monolayers, a solution of monomeric wild-type or disulphide-locked suilysin (10 µg/ml), or disulphide-locked suilysin reduced by pre-incubation with 10 mM DTT for 10 min, was overlaid with 1 µl of chloroform solution of the lipid mixture described above, at 1 mg/ml, for 25 min at 37°C and the monolayers were transferred to EM grids, as described before (*Dang et al., 2005*). To image pores on liposomes, 1 µl of a 0.3–0.5 mg/ml solution of wild-type suilysin was incubated with 1 µl of liposomes for 10 min at 37°C. Samples were negatively stained with 2% wt/vol uranyl acetate and imaged on a Tecnai F20 FEG microscope (FEI, Hillsboro, OR) at 200 kV under low dose conditions. Images were taken with a defocus of 0.5 µm on a Gatan 4k × 4k CCD camera giving a final pixel size of 1.85 Å.

### Cryo-EM

For 3D reconstructions of the prepore and the pore, 1 µl of 0.3–0.5 mg/ml solution of either wild-type or disulphide-locked suilysin was incubated with 1 µl of liposomes for 10 min at 37°C. Liposomes were then applied to lacey carbon-coated copper grids (Agar Scientific, Stansted, United Kingdom) and frozen using Vitrobot Mk3 (FEI) at 22°C and 100% humidity. Images were collected on a Tecnai G2 Polara microscope (FEI) at 300 kV, on a Gatan 4k × 4k CCD camera giving a final pixel size of 2 Å, at an electron dose of 20–25 e/Å$^2$.

## Image processing of suilysin prepores and pores on lipid monolayers

Ring images were centered and analysed by multivariate statistical analysis (MSA; *van Heel, 1984*) for classification into subsets of homogeneous diameter and subunit number. Suilysin arc length distributions were determined from negative-stain EM images at 1.85 Å pixel size and AFM images acquired at 26.8 Å pixel size. Using DNA Trace software (*Mikhaylov et al., 2013*), individual suilysin arcs were manually traced with a step size of 25 Å for both EM and AFM images. The number of monomers within each arc was then calculated by dividing the manually traced arc length by the average size of a monomer in the prepore and pore states, 23.6 Å and 25.7 Å, respectively, as estimated from rotationally averaged negative-stain EM images. This approach yielded an error within ±2 monomers as estimated from averages of rings from the EM monolayer data.

## 3D reconstruction of suilysin prepores and pores on liposomes

The defocus of the cryo-EM images was determined by CTFFIND3 (*Mindell and Grigorieff, 2003*) and phases were corrected using SPIDER (*Frank et al., 1996*). Side-view images of prepores (1374) and pores (2700) were extracted using Boxer (EMAN 1.9; *Ludtke et al., 1999*). Images were aligned in SPIDER to reprojections of pneumolysin prepore and pore maps (*Tilley et al., 2005*) and the aligned images were sorted by diameter with MSA. Initial reconstructions were calculated by back-projection of either of class sums or aligned raw images, up to 35° from the side view plane, and symmetry estimated by maximising density variance within the maps. These estimates were consistent with the outcomes of statistical analysis for negatively stained prepores and pores formed on lipid monolayers. Most of the pores (~60%) and prepores exhibited 37-fold symmetry. These 37-fold maps were further refined by projection matching with up to 20° out-of-plane tilt. MSA was used to detect and correct for misalignments. Reconstructions were calculated by back-projection in SPIDER. 450 prepore and 600 pore views were selected for the final reconstructions. The final resolution was estimated by 0.5 FSC (*Figure 1—figure supplement 2*).

## Atomic structure modelling

### Pore map fitting

First, nine different β-barrel models (corresponding to domain 3, residue range 176–225 and 272–346) with architecture $S = n/2$ were generated as described in *Reboul et al., 2012*. These nine different β-barrel models were generated with slightly varying $a$ (3.48 ± 0.1 Å) and $b$ (4.83 ± 0.1 Å) bond lengths, where the value of $a$ is the distance between Cα of adjacent residues in the same β-strand and

*b* is the distance between Cα of adjacent residues in adjacent β-strands. This resulted in β-barrel models with modest variations of radius and height, in which the β-strands are tilted by 20° from the pore axis (*Murzin et al., 1994*). All the β-barrels were fitted using the Fit-in-Map tool in Chimera (*Pettersen et al., 2004*; *Goddard et al., 2007*). Among the top three best-fitting β-barrels (CCC [cross-correlation coefficient] scores 0.43–0.44, as compared to 0.28–0.40 for the rest), the barrel with the height best matching the membrane was chosen by visual inspection. Next, the missing residues in the N-terminus of the native suilysin structure (PDB:3HVN) were modelled using MODELLER (*Sali and Blundell, 1993*). Normal Mode Analysis (NMA) was used to generate rough decoys for domains 1 and 2 using the NOMAD-Ref web server (*Lindahl et al., 2006*). 50 different decoys were obtained by randomly combining amplitudes of the first 20 modes. The value of the average coordinate root mean square deviation between the native domains and the decoys was set to 5 Å.

The decoy models and the crystal structures of the individual domains were manually fitted as rigid bodies into the pore map. The best fitting model for each domain was selected by a combination of local fit quality and geometric constraints using Chimera. For domains 1 (residue 32–48, 85–175, 226–271 and 347–370) and 2 (residues 49–84 and 371–387), the best fitting models were selected based on CCC from the decoy set. For domain 4 (residues 388–497), the crystal structure was used as the best fitting model (this domain shows very little flexibility based on NMA analysis). The geometric constraints were such that domain 2 is connected to domains 1 and 4, domain 1 is connected to the β-barrel, and domain 4 sits at the membrane surface.

For each domain, 50 models were generated with MODELLER using the above corresponding best fit as a template structure to refine the stereochemistry. Then the models were evaluated using the DOPE statistical potential score (*Shen and Sali, 2006*). The top models from the individual domains were connected with MODELLER into one partial model (containing domains 1, 2, and 4—but not 3), which was then C37 symmetrized in Chimera. Next, the map was segmented around three asymmetric units of the resulting pore model. Loop refinement was performed on the loops connecting domain 1 and the corresponding strands from the barrel (only on the central asymmetric unit). The resolution was insufficient to include β5 from domain 3 in the model. The refined asymmetric unit was C37 symmetrized to give the final pore model.

## Prepore map fitting

Only domains 1 and 4 were fitted into the prepore map. The starting structure was the crystal structure of suilysin monomer. We first rigidly fitted the whole crystal structure and then deleted domain 2 and 3, as their corresponding density was not sufficiently resolved. The fit of domain 4 was further refined to improve the CCC, taking into account the position relative to the membrane. The final fits of domain 1 and 4 were C37 symmetrized to give a final (partial) prepore model without clashes between the monomers.

## Mapping of electrostatic potential and interacting residues

We calculated the electrostatic potential of domain 1 in both prepore and pore models using the APBS method (*Baker et al., 2001*; available in Chimera) and mapped it onto their molecular surface (*Figure 1—figure supplement 3A,B*). For the pore model, optimization of the side chain rotamers was done using SCWRL (*Krivov et al., 2009*) prior to the calculation of the electrostatic potential. We further analysed the contacts in adjacent monomers of domain 1 in the prepore and pore fit (*Figure 1—figure supplement 3C,D*). We considered two residues as interacting (interface residue) if their corresponding Cβ atoms are within a distance of 7 Å (*Malhotra et al., 2014*).

## AFM sample preparation

Small unilamellar vesicles were injected onto a freshly cleaved mica surface at a concentration between 5 and 25 nM in the presence of 60 μl of 20 mM Tris, 150 mM NaCl, 20 mM $MgCl_2$, pH 7.8. Incubation of the vesicles on the mica for 30 min at room temperature allowed them to rupture and adsorb onto the surface, yielding an extended lipid bilayer film. Any remaining vesicles were removed by gently rinsing with 80 μl of the adsorption buffer. The rinsing process was repeated 3–7 times to ensure a clean and uniform surface conducive for AFM imaging. Wild-type and disulphide-locked suilysin were injected into a 150 μl fluid cell containing the supported lipid bilayers and allowed to equilibrate for ~10 min prior to imaging. The concentration of suilysin in the various AFM experiments was 12–180 nM.

For the doping assays (*Figure 4—figure supplement 1*), wild-type and locked suilysin were mixed in the desired molar ratios and 60 nM of the protein mixture was incubated on the lipid bilayers for 10 min.

## AFM imaging and data processing

Real-time topographic images of suilysin on the supported lipid bilayers were collected on a Multimode 8 system (Bruker, Santa Barbara, CA) by performing rapid force-distance (PeakForce Tapping) curves. The PeakForce method continuously records force-distance curves with a user-defined force set-point (here about 50 pN) that is referenced to a continuously adjusted baseline. Typically, these force-distance curves were recorded at a frequency of 2 kHz with a maximum tip-sample separation between 5 and 20 nm. The topographic features were verified for consistency between trace and retrace images, as well as for their reproducibility in subsequent scan frames. For imaging, the vertical scan limit was reduced to ~1.5 μm. Typically, images were recorded at 0.2–1 frames/min. Suilysin pre-pores and pores were also imaged at rates of up to 10 frames/min using a home-built AFM system and miniaturized cantilevers (*Leung et al., 2012*), but this did not yield information additional to the data presented here. Suilysin prepores were only resolved at high concentration on the membrane (*Figure 2C*), or when the temperature was lowered to 15°C (*Video 1*). The real-time, low-temperature measurements were carried out on a Dimension FastScan AFM system in tapping mode with images acquired at 4 frames/min using FastScan Dx probes (Bruker).The AFM probes used for suilysin imaging had nominal spring constants ranging from 0.1 to 0.7 N/m and resonance frequencies between 10 and 130 kHz in liquid. We used silicon nitride AFM probes with batch-processed silicon tips including MSNL E and F (Bruker), ScanAsyst Fluid+ (Bruker), and cantilevers with individually grown carbon tips, for example, Biotool (Nanotools, Munich, Germany). Batches of AFM probes were screened for tip sharpness and appropriate tilt angles prior to data collection.

All AFM imaging was performed in the presence of 20 mM Tris, 150 mM NaCl, 20 mM $MgCl_2$, pH 7.8 with either an E or a J scanner with an integrated temperature control. Images were analysed by either the Nanoscope Analysis software package (Bruker) or using the open-source SPM analysis software, Gwyddion (www.sourceforge.net). The raw AFM images were plane-levelled and subsequently line-by-line flattened using the lipid membrane as reference. A Gaussian filter with a full-width-half-maximum of 2-pixels was applied to smooth out high frequency noise where necessary.

## Oligomerization model

The assembly of suilysin (SLY) in the prepore state was described by the irreversible reactions $SLY_{n-1}^{(pre)} + SLY_1^{(pre)} \rightarrow SLY_n^{(pre)}$ for oligomerization via monomer-association with a rate constant $k_a$. Here $n$ denotes the number of monomeric subunits in an oligomer, ranging from 1 to the maximum number of monomers in a complete ring, $N = 37$. The prepore monomers originated from the binding of soluble suilysin monomers to the membrane, $SLY_1^{(sol)} \rightarrow SLY_1^{(pre)}$, here assumed to occur with a rate constant $k_b$.

$\sigma_n(t)$ was defined as the number of suilysin prepore $n$-mers per unit area on the membrane, and $C$ as the number of monomers in solution above a unit membrane area, immediately after injection of suilysin at time $t = 0$. With these definitions, the oligomerization reactions can be modelled by the rate equations

$$\frac{d\sigma_n(t)}{dt} = \delta_{n,1} k_b C e^{-k_b t} + \sum_{m=1}^{n-1} \frac{1}{2} k_a \left( \delta_{m,1} + \delta_{n-m,1} \right) \sigma_m(t) \sigma_{n-m}(t)$$

$$- \sum_{m=1}^{N-n} k_a \left( \delta_{m,1} + \delta_{n,1} \right) \sigma_m(t) \sigma_n(t).$$

These reactions lead to kinetically trapped prepore assemblies on depletion of free monomers on the membrane, that is, when $\sigma_1(t) \rightarrow 0$ as time evolves. With the substitutions $t = \tau/k_b$ and $\sigma_n(t) = s_n(\tau)k_b/k_a$, the rate equations can be rewritten in terms of a dimensionless surface density $s_n(\tau)$ and a dimensionless time $\tau$, to demonstrate that the shape of the solution for $s_n(\tau \rightarrow \infty)$ versus $n$, and thus of the resulting arc length distribution ($\sigma_n(t \rightarrow \infty)$ versus $n$), is a function of the parameter $Ck_a/k_b$ only.

The coupled and nonlinear differential equations for $s_n(\tau)$ were integrated numerically for different $Ck_a/k_b$ using the Runge-Kutta method, until a stationary solution was reached. For fitting experimental data, $C$ was determined from the accumulated length of all measured oligomers, normalized to the measured membrane area. The best $k_a/k_b$ then followed from the numerical solution that yielded the lowest sum of squared residues.

## Acknowledgements

This work has been funded by the BBSRC (BB/G011729/1, BB/J005932/1, BB/J006254 and BB/K01692X/1), the ERC (advanced grant 294408), and the Leverhulme Trust (RPG-2012-519). We thank Richard Thorogate for technical support, Dan Clare, Elena Orlova, and Luchun Wang for help with EM and image processing, Dave Houldershaw and Alan Lowe for computing support, Chanmin Su and Khaled Kaja (Bruker) for AFM probes and access to equipment, and James Whisstock for comments on the manuscript.

Accession codes for EM maps: prepore EMD-12698, pore EMD-12711.

## Additional information

### Funding

| Funder | Grant reference number | Author |
| --- | --- | --- |
| Biotechnology and Biological Sciences Research Council | BB/G011729/1, BB/J005932/1, BB/J006254 and BB/K01692X/1 | Helen R Saibil, Bart W Hoogenboom |
| European Research Council | 294408 | Helen R Saibil |
| Leverhulme Trust | RPG-2012-519 | Maya Topf |

The funders had no role in study design, data collection and interpretation, or the decision to submit the work for publication.

### Author contributions

CL, NVD, Conception and design, Acquisition of data, Analysis and interpretation of data, Drafting or revising the article; NL, BWH, Conception and design, Analysis and interpretation of data, Drafting or revising the article; AWH, Acquisition of data, Analysis and interpretation of data, Drafting or revising the article; IF, APP, CFR, MAD, Development of atomic models, Analysis and interpretation of data, Drafting or revising the article; NJ, MPD, Expression, Purification, Testing of wild-type and mutant proteins, Drafting or revising the article, Contributed unpublished essential data or reagents; DO, Development of oligomerization model, Analysis and interpretation of data, Drafting or revising the article; PWA, RL, Conception and design, Drafting or revising the article, Contributed unpublished essential data or reagents; MT, Analysis and interpretation of data, Drafting or revising the article; HRS, Conception and design, Analysis and interpretation of data, Drafting or revising the article, Contributed unpublished essential data or reagents

## Additional files

### Major datasets

The following datasets were generated:

| Author(s) | Year | Dataset title | Dataset ID and/or URL | Database, license, and accessibility information |
| --- | --- | --- | --- | --- |
| Dudkina NV, Leung C, Lukoyanova N, Hodel AW, Farabella I, Pandurangan AP, Jahan N, Damaso MP, Osmanovic D, Reboul C, Dunstone MA, Topf M, Andrew PW, Lonnen R, Saibil HR, Hoogenboom BW | 2014 | Cryo EM structure of suilysin prepore | EMD-12698 | Publicly available at EMDataBank. |
| Dudkina NV, Leung C, Lukoyanova N, Hodel AW, Farabella I, Pandurangan AP, Jahan N, Damaso MP, Osmanovic D, Reboul C, Dunstone MA, Topf M, Andrew PW, Lonnen R, Saibil HR, Hoogenboom BW | 2014 | Cryo EM structure of suilysin pore | EMD-12711 | Publicly available at EMDataBank. |

The following previously published dataset was used:

| Author(s) | Year | Dataset title | Dataset ID and/or URL | Database, license, and accessibility information |
|---|---|---|---|---|
| Xu L, Huang B, Du H, Zhang CX, Xu J, Li X, Rao Z | 2010 | Crystal structure of cytotoxin protein suilysin from *Streptococcus* suis | http://www.pdb.org/pdb/explore/explore.do?structureId=3hvn | Publicly available at RCSB Protein Data Bank. |

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
