## [Decision Letter]

Thank you for sending your work entitled “Stepwise visualization of membrane pore
formation by suilysin, a bacterial cholesterol-dependent cytolysin” for
consideration at *eLife*. Your article has been favorably evaluated by
John Kuriyan (Senior editor) and 3 reviewers, one of whom is a member of our Board of
Reviewing Editors.

The following individuals responsible for the peer review of your submission have agreed
to reveal their identity: Volker Dötsch as Reviewing editor and Andreas Engel as
one of the other two reviewers.

The Reviewing editor and the other reviewers discussed their comments before we reached
this decision, and the Reviewing editor has assembled the following comments to help you
prepare a revised submission.

Carl Leung and colleagues present a structural investigation of the pore forming protein
suilysin. They used AFM and cryo-EM to investigate both, the structure and dynamics of
pore formation. Interpretation of these data was done by fitting and modeling. The
crystal structure of suilysin reveals key-shaped molecule in its soluble form, which
undergoes a conformational change when binding to the target membrane. Cryo- and
negative stain EM of cys-locked suilysin-prepores and wt-pores revealed rings of
different diameters but identical number (37) of subunits, supporting the conformational
change. A 3D map was reconstructed from cryo-EM projections of vitrified rings.

While these results were seen as interesting by all reviewers, some discussion about the
impact of these findings arose. In particular two questions were discussed:

1) The first one focused on the observation that lipids are released upon insertion.
This issue has been discussed in the field for quite a while but has not been
convincingly shown. The interpretation that this release occurs is based on the
observation of density in the time resolved AFM images that appear and then disappear
again. Showing conclusively that this density is indeed lipids would be a significant
step ahead. Is there additional experimental evidence that this interpretation is
correct? Could the authors for example provide a statistical evaluation of the size of
these densities and explore if this is consistent with the expected size of the released
lipids?

In addition, the process of when and how lipid ejection occurs should be discussed.

2) The provided structural model is of higher resolution than all other models that have
been published before. Is there experimental evidence from mutational analysis that the
structural model is correct? Again any evidence that the proposed interfaces indeed play
a role during this process would significantly strengthen the impact of these
results.

Further issues that should be addressed are:

3) “Negative stain EM and rotational symmetry analysis of complete rings of
disulphide locked (Gly52Cys/Ser187Cys) suilysin prepores and wild-type pores formed on
lipid monolayers revealed that most rings contain 37 subunits.” This is not
documented properly: it should be, at least in the supplemental data. Cryo-EM reaches a
resolution of 15 Å where the subunits are just visible. Negative stain is not
likely to produce the same resolution; usually 25 Å is good. So how sure then is
the 37-fold symmetry?

4) “The pore structure is similar to that observed with pneumolysin, but with
improved resolution, resolving the β-barrel and defining hinge bending and domain
movements.” There is no evidence that the β-barrel and defining hinge
bending are resolved in the 15 Å map. It is that a convincing model has been
developed to demonstrate this. Convincing is the height change seen by AFM - the 2
observation together make this work exciting. Please reformulate.

5) In the title and in some parts of the manuscript suilysin is named as a
cholesterol-dependent cytolysin. Does cholesterol play a role in the assembly and
insertion process? What is the role of cholesterol?

6) The attachment to the membrane is assumed to be irreversible. What is the basis for
this assumption? Is there a conformational change that occurs upon membrane binding? And
a second upon oligomerization followed by the final conformational change when the pore
is formed?

7) The assembly itself is assumed not to be cooperative. What about the conformational
change from the pre-pore to the pore? Is this a cooperative process?

---

## [Author Response]

*1) The first one focused on the observation that lipids are released upon
insertion. This issue has been discussed in the field for quite a while but has not
been convincingly shown. The interpretation that this release occurs is based on the
observation of density in the time resolved AFM images that appear and then disappear
again. Showing conclusively that this density is indeed lipids would be a significant
step ahead. Is there additional experimental evidence that this interpretation is
correct? Could the authors for example provide a statistical evaluation of the size
of these densities and explore if this is consistent with the expected size of the
released lipids*?

*In addition, the process of when and how lipid ejection occurs should be
discussed*.

The conclusion that lipids are ejected is based on the appearance and subsequent
disappearance of density in the real-time AFM images, as pointed out by the reviewers,
but also on the time at which this occurs. These observations are made within minutes
after injection of DTT to trigger the prepore-to-pore transition, and just before the
first detection of membrane lesions within the pores.

As additional experimental evidence, we have now included a new Figure 3—figure supplement 1 that demonstrates that
appearance of additional density is reproducible, after triggering the prepore-to-pore
transition, in an independent time-resolved experiment carried out along the same lines
as the one reported in Figure 3. It can thus be
excluded that the additional density is artefactual. This leaves only two possible
candidates to explain the additional high features appearing in Figure 3 (and in Figure 3—figure supplement 1): suilysin or lipids. While we cannot exclude the
ejection of some protein, our interpretation of ejected lipids is supported (i) by the
subsequent appearance of collapsed lipid vesicles on the membrane (plateaus in Figure 3), and (ii) by the fact that after
completion of pore formation, there are obviously lipids missing within the pore lumens
(see holes in membrane, Figure 3), as described
in the manuscript.

We appreciate the suggestion of a statistical size evaluation in the reviewers’
comment above, but would like to point out that for such soft and clearly mobile
features in AFM images (see streaky appearance of the white features in Figure 3), reliable and accurate volume
measurements are nigh impossible. For completeness, we have now also attempted confocal
fluorescence and TIRF microscopy experiments with fluorescent lipids, but unfortunately
with inconclusive results.

In the Discussion section, in the paragraph “Regardless of the extent of
oligomerization…” we have specified that the lipid ejection occurs upon
conversion of suilysin from its prepore to pore state, hence upon the unfurling of the
β-hairpins into the membrane. We have now expanded the corresponding paragraph to
briefly discuss how lipid ejection may occur, specifically: “The results imply
that the hydrophilic inner surface of the partially or fully completed β-barrel
is sufficient to destabilize the lipid membrane in the pore lumen, leading to ejection
of lipid micelles from the pore.”

*2) The provided structural model is of higher resolution than all other models
that have been published before. Is there experimental evidence from mutational
analysis that the structural model is correct? Again any evidence that the proposed
interfaces indeed play a role during this process would significantly strengthen the
impact of these results*.

In response to this question, we have analysed the electrostatic properties of the
surfaces of domain 1 that form the major interfaces in the prepore and pore oligomers.
The results show that both models have extended regions of complementary charge on the
predicted interacting surfaces. Although the extent of complementary charge is less in
the pore model, in this case the oligomer is stabilized by the beta barrel of the pore.
This analysis lends support to both models, is discussed in the Results section on the
Cryo-EM data, and is shown in the new Figure 1—figure supplement 3.

*Further issues that should be addressed are*:

*3) “Negative stain EM and rotational symmetry analysis of complete rings
of disulphide locked (Gly52Cys/Ser187Cys) suilysin prepores and wild-type pores
formed on lipid monolayers revealed that most rings contain 37 subunits.” This
is not documented properly: it should be, at least in the supplemental data. Cryo-EM
reaches a resolution of 15 Å where the subunits are just visible. Negative stain
is not likely to produce the same resolution; usually 25 Å is good. So how sure
then is the 37-fold symmetry*?

The subunit repeat is clearly resolved around the outer edge of the ring in negative
stain end views on membrane monolayers. End view class averages and rotational
correlation plots documenting the presence of 36-39-mers are presented in a new Figure 1—figure supplement 1.

*4) “The pore structure is similar to that observed with pneumolysin, but
with improved resolution, resolving the β-barrel and defining hinge bending
and domain movements.” There is no evidence that the β-barrel and
defining hinge bending are resolved in the 15 Å map. It is that a convincing
model has been developed to demonstrate this. Convincing is the height change seen by
AFM - the 2 observation together make this work exciting. Please
reformulate*.

This sentence has now been reformulated to: “The pore structure is similar to
that observed with pneumolysin, but with improved resolution, and showing significant
differences in the hinge bending and domain movements.”

*5) In the title and in some parts of the manuscript suilysin is named as a
cholesterol-dependent cytolysin. Does cholesterol play a role in the assembly and
insertion process? What is the role of cholesterol*?

We have now specified in the introduction that CDCs, exemplified by perfringolysin O,
require >∼30% for membrane binding (with the new reference [12]). Consistent with that
observation on perfringolysin O, our experiments with suilysin on membranes with 10% and
20% cholesterol content did not show any significant sign of membrane binding and pore
formation (data not shown, since this point is well established in the CDC literature).
As for the dependency on cholesterol above the ∼30% threshold, we acquired data
mostly at 33% and 55% cholesterol content, but did not observe large changes in
oligomeric populations in this range (compare, e.g., Figure 4 and Figure 4—figure supplement 1).

*6) The attachment to the membrane is assumed to be irreversible. What is the
basis for this assumption? Is there a conformational change that occurs upon membrane
binding? And a second upon oligomerization followed by the final conformational
change when the pore is formed*?

Firstly, the assumption of irreversible oligomerization is dictated by the presence of a
broad maximum for incomplete (but oligomeric) assemblies in the prepore state (Figure 4): Because the chemical interface between
all subunits is identical, this maximum cannot be explained as an equilibrium result,
and must thus represent kinetically trapped assemblies, hence irreversible
oligomerization.

As for monomer attachment to the membrane, it will eventually become irreversible
because of the irreversibility of the subsequent oligomerization. The monomer attachment
to the membrane can therefore be approximately described by an effective irreversible
rate equation. The above-mentioned assumption of irreversible membrane binding was thus
made because it simplifies the model (reducing the number of free parameters) while it
is not critical for the modelling results.

*7) The assembly itself is assumed not to be cooperative. What about the
conformational change from the pre-pore to the pore? Is this a cooperative
process*?

Our experiments in Figure 4—figure supplement 3 suggest that the prepore-to-pore transition is indeed a cooperative process,
as we have now made more explicit in the Discussion (“The activity of wild-type
suilysin was greatly impaired…”).